# Generalized Reputation Computation Ontology and Temporal Graph Architecture

**Anton Kolonin** *
The Artificial Intelligence Research Center
Novosibirsk State University
Pirogova 1, Novosibirsk, 630090, Russia
`akolonin@gmail.com`

## Abstract

The problem of reliable democratic governance is important for survival of any community, and it will be more critical over time communities with levels of social connectivity in society rapidly increasing with speeds and scales of electronic communication. In order to face such challenge, different sorts of rating and reputation systems are being developed, however reputation gaming and manipulation in such systems appears to be serious problem. We are considering use of advanced reputation system supporting "liquid democracy" principle with generalized design and underlying ontology fitting different sorts of environments such as social networks, financial ecosystems and marketplaces. The suggested system is based on "temporal weighted liquid rank" algorithm employing different sorts of explicit and implicit ratings being exchanged by members of the society. For the purpose, we suggest "incremental reputation" design and graph database used for implementation of the system. Finally, we present evaluation of the system against real social network and financial blockchain data. The entire framework is expected to be the foundation of any multi-agent AI framework, so the evolution of distributed multi-agent AI architecture and dynamics will be based on the organic reputation scores earned by the agents that are part of it.

## 1 Introduction

The entire history of human communities shows that no reliable solution for reaching truly long-term democratic consensus in society has been invented so far Hazin & Shheglov (2018). Different sorts of social organizations and governance policy have been tried, starting with completely centralized governance in from of "monarchy" and ending with completely distributed "anarchy". In any case, the crucial part of social organization design remains principles of reaching social consensus recognized and accepted by entire society.

One form of consensus is known to be based on brute force in animal groups and ancient societies, serving the minority having the access to the force. The same solution is reproduced in nowadays distributed computing systems such as blockchains and called Proof-of-Work (PoW). More advanced form of consensus employed by human race now is based of financial capabilities of members of society. It is known to lead to the situation when "reacher become richer" and gain more and more power. This is is also replicated with the same phenomena observed in latest developments of blockchain systems relying on Proof-of-Stake (PoS). Further, in some of the latest blockchain systems, the employed solution is so-called Delegated Proof-of-Stake (DPoS). The latter solution still implies the rule on basis of financial capabilities being implemented indirectly, by means of manually controlled voting process to selected delegates who conduct the governance of the system.

The limited list of the options above leads to the situation that consensus in any community or a distributed computing system may be easily taken over by a group which concentrates substantial amount of power (be it physical, military or computational one) of financial resources. Obviously, the latter group may be minority organized towards the goals hostile to the majority of the commu-

---

*GitHub: `https://github.com/aigents`, `https://github.com/akolonin`

nity. So, the better and more reliable and fair forms of consensus are in demand. The one of them is so-called Proof-of-Reputation (PoR) Kolonin et al. (2018) consensus being discussed further.

## 2 REPUTATION SYSTEM CONCEPT

The Reputation Consensus suggested earlier in Kolonin et al. (2018) is implementing Proof-of-Reputation (PoR) principle, opposing power of brute force (PoW) and power of money (PoS or DPoS), as shown on Fig. 1. It is anticipated that the Proof-of-Reputation can make it possible to implement system of Liquid Democracy to cure known issues of representative democracy, influenced by power of money. In this case the governing power of member of a human or artificial society depends on Reputation of the member earned on basis of the following principles.

1) Reputation may be computed by means of different measures, called "ratings", performed explicitly or implicitly by all members of community, called "raters", in respect to ones who Reputation is being computed for, called "ratees", with account to Reputations of the "raters" themselves.

2) Reputation computation is time-scoped, so that measures collected by a ratee in the past are less contributing to its current Reputation than the latest ones, which have more impact.

3) Data used to compute the Reputations of all raters and ratees with the ratings that they issue or receive is open so that audit of Reputations and the historical measures over the history can be performed in order to prevent Reputation manipulation and gaming.

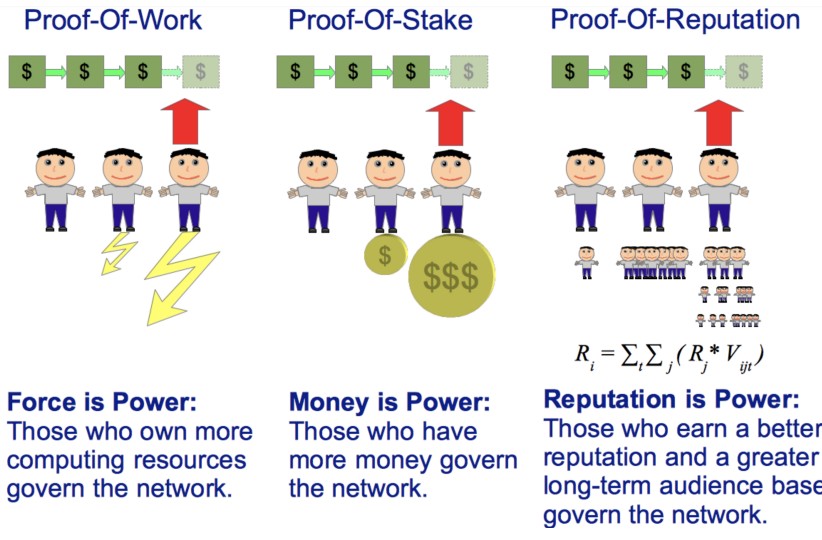

Figure 1: Types of consensus in distributed systems such as Proof-of-Work, Proof-of-Stake and Proof-of-Reputation.

There may be many kinds of explicit and implicit measures contributing to evaluation of Reputation considered, depending on a kind of practical use case and implementation of a given Reputation system Swamynathan et al. (2010), Sänger & Pernul (2018), Androulaki et al. (2008), Gupta et al. (2003), Blömer et al. (2014). For instance, it could be system serving social network Kolonin (2019) or a marketplace Kolonin et al. (2019).

Applicability of the measures or ratings may depend on accuracy and reliability that they may provide as well as resistance to attack vectors targeting takeover of the consensus by means of reputation cheating and gaming for specific case. In the current work we are trying to come up with generic purpose architecture so are going to enumerate all possible options. Respectively, following the work Kolonin et al. (2018) we consider such measures as: a) members explicitly staking financial values on other members; b) members explicitly providing ratings in respect to transactions committed with other members; c) implicit ratings computed from the financial values of transactions between the members; d) evaluation of textual, audial and video reviews or mentions made by members in respect to other members or transactions between them.

## 3  REPUTATION SYSTEM ONTOLOGY

Generic purpose ontology for a Reputation System serving an online community may be depicted on Fig. 2 with the following entities identified (some of them show on the Fig. 2 and some omitted).

1) Account – primary entity of a Reputation System to play a "rater" or "ratee" role or both, can be impersonating a physical person, business or governmental entity, robotic system etc.;

2) Smart Contract – secondary entity specific to blockchain environments which may belong to some of the Accounts;

3) Product/Service – secondary entity specific to marketplace environments identifying products or services provided by some of the Accounts;

4) Post/Comment – secondary entity specific to social networks or messaging environments representing any sort of textual, audial or any other sensible non-financial communication;

5) Word – tertiary entity specific to social networks or messaging environments identifying a word used in a post/comment and carrying positive or negative sentiment which can be purposed to assess its impact on reputation;

6) Tag/Category – any classification of any of the entities identified above.

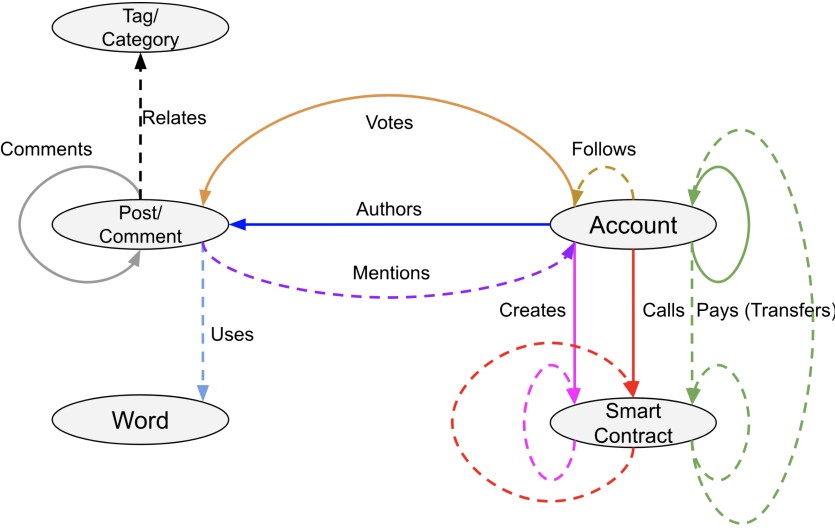

Figure 2: Simplified ontology of the generic-purpose Reputation System for online environments.

Besides the entities above, there are relationship connecting them, such as the following.

1) Votes – any "Vote", "Like" or "Rate" event providing assessment of any communication on behalf of a "rater" Account;

2) Authors – authorship of a communication;

3) Mentions – mentioning of an Account in a communication;

4) Uses – use of a Word in a communication;

5) Relates – relevance of a Post/Comment or a Product/Service (the latter is not shown on  2) to specific Tag/Category;

6) Creates – creation of a Smart Contract by an Account;

7) Calls – call of a Smart Contract by an Account;

8) Pays (Transfers) – transfer of a funds for a Service/Product (not shown on Fig. 1), which may be involving Smart contract or not, assuming that each Pay or Transfer may also have multiple

properties such as amount of the payment and currency, inventory of services or products being ordered and paid and the optional rating "Rate" event, if applies.

9) Follows – following of one Account by another in a social network;

10) Provides – provision of a Product/Service (not shown on Fig. 2) on behalf of an Account;

## 4 COMPUTATIONAL MODEL

According to Kolonin et al. (2018), the reputation $R_i(t)$ of society member $i$ at moment $t$ can be computed incrementally on the basis of its own reputation at the previous moment $R_i(t-1)$, and some default reputation $R_d$ taken as initial default value. Changes in the reputation of $i$ can be caused by different sorts of ratings issued by multiple other members $j$, in respect to a particular aspect of reputations $k$ and specific domain category of reputation $c$. The aspect $k$ is assumed to be a generic measure like reliability, quality or timeliness while the category c may identify an area of a member's expertise such as painting, stock prediction or pizza delivery.

The ratings based on the relationships identified in the previous secion can be divided into two types. First, there are "static" (called "endorsing" in Kolonin et al. (2018)) ratings $S_{ijkc}$, like "Follows" or "Provides" above, which may be or may not be be present at any time $t$, being granted or revoked from $j$ to $i$. Next, there are transactional ratings $F_{ijkce}$ that can be recorded in a history of interactions, being associated with either financial transactions from $j$ to $i$ (financial ratings such as "Pays" above) or acts of voting (voting ratings such as "Votes" above) in respect to particular events $e(t)$, such as publications, posts, comments, nominations or tasks and duties being served by $i$ in respect to $j$ (such as "Authors" above). Most ratings can be either explicit or implicit. Explicit voting ratings come with rank value expressed as positive, negative or any number at some scale, while implicit ones are comments and reviews authored by $j$ in respect to $i$ where actual value or rating should be somehow from the media used for comment or review, such as natural language text. Static (endorsing) ratings $S_{ijkc}$ may be backed up with financial stake value $Q_{ij}$. Transactional voting ratings $F_{ijkce}$ can be backed up by a financial value $G_{ije}$. For example, the value of a customer's $j$ vote in respect to the quality of service provider $i$ may be weighted with account of cost of the entire service $e(t)$.

The rating values maybe scaled in the range $-1$ to $1$ for negative and positive ratings, while for presentation purposes they may be scaled to $-5$ to $5$, $0$ to $10$ or whatever seems visually intuitive. For financial ratings, experimentation with Ethereum blockchain, has shown that it is desirable to normalize the nonlinear distributions of financial values of transactions as follows:

$$F'_{ijkce} = \frac{log_{10}(F_{ijkce})}{MAX(log_{10}(F_{ijkce}))}$$

Ratings for different aspects $k$ can be blended to infer overall reputation using a system-wide blending parameter $H_k$. Then, the following formulae can identify differential reputation at time tn as a relative increase of reputation due to endorsing $dS_i(t_{n-1}, t_n)$ and transactional $dF_i(t_{n-1}, t_n)$ components, with $t$ for events $e(t)$ varying in range from $t_{n-1}$ to $t_n$.

$$dS_i(t_{n-1}, t_n) = \frac{\sum_k(H_k * \frac{\sum_{jct}(S_{ijkc}(t_n)*Q_{ijc}(t_n)*R_j(t_{n-1})))}{\sum_{jct}(Q_{ijc}(t_n)*R_j(t_{n-1}))}}{\sum_k(H_k)}$$

$$dF_i(t_{n-1}, t_n) = \frac{\sum_k(H_k * \frac{\sum_{jct}(F_{ijkce}(t)*G_{ijce}(t)*R_j(t_{n-1}))}{\sum_{jct}(G_{ijce}(t)*R_j(t_{n-1}))}}{\sum_k(H_k))}$$

In simplified form, when no aspects or categories are considered, increases of endorsing and transactional reputations can be simplified as follows.

$$dS_i(t_{n-1}, t_n) = \frac{\sum_{jt}(S_{ij}(t_n) * Q_{ij}(t_n) * R_j(t_{n-1}))}{\sum_{jt}(Q_{ij}(t_n) * R_j(t_{n-1}))}$$

$$dF_i(t_{n-1}, t_n) = \frac{\sum_{jt}(F_{ije}(t) * G_{ije}(t) * R_j(t_{n-1}))}{\sum_{jct}(G_{ije}(t) * R_j(t_{n-1}))}$$

In practical implementation, either endorsing or transactional reputation can be used. In case of using both, a blended increase of reputation may be computed with blending factors $S$ and $F$ for each of the two reputations, respectively.

$$dP_i(t_{n-1}, t_n) = \frac{(S * dS_i(t_{n-1}, t_n) + F * dF_i(t_{n-1}, t_n))}{(S + F)}$$

Differential reputation can be further normalized by a maximum absolute reputation increase per time step:

$$P_i(t_{n-1}, t_n) = \frac{dP_i(t_{n-1}, t_n)}{MAX_i(ABS(dP_i(t_{n-1}, t_n)))}$$

Based on reputation earned in the previous period from to to tn-1, the new reputation for latest time tn can be computed by blending the previous value with the differential one.

$$R_i(t_n) = \frac{((t_{n-1} - t_o) * R_i(t_{n-1}) + (t_n - t_{n-1}) * P_i(t_{n-1}, t_n))}{(t_n - t_o)}$$

As it has been discovered in experiments discussed in Kolonin et al. (2018), a linear computation of reputation applied to experimental communities results in a quite nonlinear distribution of reputation values in the community, where very few members have very high values, but the rest of the community have reputations equal to zero. To improve the distribution for practical purposes, nonnegative logarithmic differential reputation can be computed as follows, so the $lP_i(t_{n-1}, t_n)$ can be used instead of $dP_i(t_{n-1}, t_n)$ in the two formulae above.

$$lP_i(t_{n-1}, t_n) = SIGN(dP_i(t_{n-1}, t_n)) * log_{10}(1 + ABS(dP_i(t_{n-1}, t_n)))$$

The reputation evaluation framework presented in Kolonin et al. (2018) can be modified to allow earned reputation decay more quickly or slowly. We can apply extra blending factors to the most recent time interval importance and to earlier time intervals when computing $R_i(t_n)$, so that previously earned reputation values can decay faster or slower after being amended with the latest differential reputation.

It is also possible to compute more fine-grained reputations specific to different aspects or categories, as we will show for transactional differential reputation below. Based on these ideas, more specific reputations $R_{ic}(t_n)$, $R_{ik}(t_n)$ and $R_{ikc}(t_n)$ can be computed within the community according to the following formulas.

$$dF_{ic}(t_{n-1}, t_n) = \frac{\sum_k (H_k * \frac{\sum_{jt}(F_{ijkce}(t) * G_{ijce}(t) * R_j(t_{n-1}))}{\sum_{jt}(G_{ijce}(t) * R_j(t_{n-1}))})}{\sum_k (H_k)}$$

$$dF_{ik}(t_{n-1}, t_n) = \frac{\sum_{jct}(F_{ijkce}(t) * G_{ijce}(t) * R_j(t_{n-1}))}{\sum_{jct}(G_{ijce}(t) * R_j(t_{n-1}))}$$

$$dF_{ikc}(t_{n-1}, t_n) = \frac{\sum_{jt}(F_{ijkce}(t) * G_{ijce}(t) * R_j(t_{n-1}))}{\sum_{jt}(G_{ijce}(t) * R_j(t_{n-1}))}$$

The computational framework suggested above, according to Kolonin et al. (2018), can be designed and implemented in many possible ways, based on decisions made in respect to temporal scoping of the reputation calculation and its maintenance and storage options, as discussed further on. In the end, we introduce notions of "Reputation consensus", "Proof-of-Reputation" and "Reputation mining".

Run-time performance of reputation system and its computational cost would depend on time scoping, based on interval spanning between cycles of reputation evaluations between times $t_{n-1}$ and $t_n$.

On one end, there is "lifetime" recalculation where all ratings between $t_0$ and $t_n$ are counted. In this case, it is possible to account for backdated changes in the ratings history to be accounted for with later re-calculation. However, this is much more expensive and time consuming. Also, in this case reputation decay can not be achieved as designed above and complication of differential reputation functions are required, introducing an extra time-bound weighting function which would give higher weights for more recent ratings.

On the other end, there is "incremental" recalculation with time intervals between $t_0$ and $t_n$ corresponding to intervals between subsequent transactions, so every transaction effects in global change of reputation. No reputation change delay may be experienced in this case yet implementation of this in a distributed way may get to be not trivial. At the same time, it might be beneficial for distributed systems not based on the blockchain.

In between the two above, there is "up-to-date" recalculation with time intervals between $t_0$ and $t_n$ being such as years, quarters, months, weeks, days etc. This would be more efficient and fast however reputations change may be delayed, getting outdated closer to the end of recalculation interval. Finally, there is a hybrid between the last two such as "blocked incremental" recalculation where blocks of latest subsequent transactions are used to identify the time interval. It might be beneficial to have this implemented in distributed blockchain systems.

The computational model above extends the basic "liquid rank" concept suggested by Brin & Page (1998) in two ways. First, it adds multiple aspects of the ratings beyond simple linking, so the relationships with different meanings may be assigned different "weights", plus the relationship on itself may be given a "weight", such as value of financial transaction or number of words in the comment to one's post, so it can be called "weighted liquid rank", according to Kolonin et al. (2022). Second, adding a time dimension to it, so that the raw ratings supporting the reputation ranks can be constrained by time ranges, allows it to be called a "temporal weighted liquid rank".

## 5 INCREMENTAL IMPLEMENTATION AND TEMPORAL GRAPH DATABASE

Given the high volumes of data being involved in real-world financial networks and requirement for low response time on behalf of production systems, we focus on "incremental" design option for Reputation System described in the earlier work Kolonin et al. (2018).

The implementation employs in-memory graph database of Aigents project available in Java as open source `https://github.com/aigents/aigents-java`.

The first key feature of the Aigents graph database is its ability to store labeled temporal graphs with possibility to attach any value to an edge between two vertices, so the value can be either numeric value indicating weight or strength of the relationship in a weighted graph or a compound truth value in probabilistic logic network Goertzel et al. (2008). In case of the Reputation System implementation, it may keep either single rating assessment or a financial transaction value and currency, or the combination of them all so he ratings may be weighted by financial values as presented in the earlier work Kolonin et al. (2019). Notably, the edge may contain entire probabilistic distribution or the list of associated transactions along with ratings and financial values.

The second feature of the Aigents graph database is its ability to slice graphs on temporal basis so that each time period is stored in separate subgraph while the subgraphs can be arbitrary merged, or the subgraphs of multiple temporal subgraphs can be extracted and merged over time. In the case of Reputation System implementation, it has appeared logical to keep all transactions segmented in temporal subgraphs specific to what is called recalculation period Kolonin et al. (2018) or observation period according to Kolonin et al. (2019), which is 1 day by default in the current implementation.

The indexing of the data is primarily temporal and secondary based on vertices and types of the relationships. During the run-time, temporal ranges of subgraphs being processed on time scale are bound to memory resources available as well, so that amount of storage space limits time range that can be processed simultaneously. However, the incremental nature of the reputation recalculation given the "incremental" design option needs only few subgraphs to be present in memory at one time.The indexing of the data is primarily temporal and secondary based on vertices and types of the relationships. During the run-time, temporal ranges of subgraphs being processed on time scale are bound to memory resources available as well, so that amount of storage space limits time range that can be processed simultaneously. However, the incremental nature of the reputation recalculation given the "incremental" design option needs only few subgraphs to be present in memory at one time.

Specifically, the two types of graphs are purposed for the Reputation System implementation. First, there is "reputation evidence data" with historical data accumulated for each of the observation

period such as single day by default. Second, there is "reputation state" graph keeping current "reputation balance" for each of the periods. That is, for each of the reputation update process per observation period accordingly to the algorithm specification Kolonin et al. (2019), the only three subgraphs should be present in physical memory – one "reputation evidence data" for the current period and two "reputation state" subgraphs for current and previous periods.

## 6    Application to real Social Network and Blockchain Data

The illustration of how the Reputation System can work has been evaluated with use of real world data extracted from public blockchain Steemit as discussed in the earlier work Kolonin (2019). We have tried to evaluate how the level of reputation computed by the Reputation System for participants of Steemit social network corresponds to the evaluations of trust and credibility given to them manually. For the purpose, we were computing reputations for entire network for long period of time based on both social and financial data involving voting for posts, commenting on posts and sending financial payments as well. Further, the computed reputation ranks were compared against the "black lists" maintained by network administrators and volunteers as well as against the lists of "whales", called so for listing well-known publicly available participants. The extra study has been performed to see how the reputation changes over time for accounts of different kinds ("black-listed" or "whales"), assuming every account starts with default reputation of 0.5 which may get changed to higher or lower over time, as it is shown on Fig. 3.

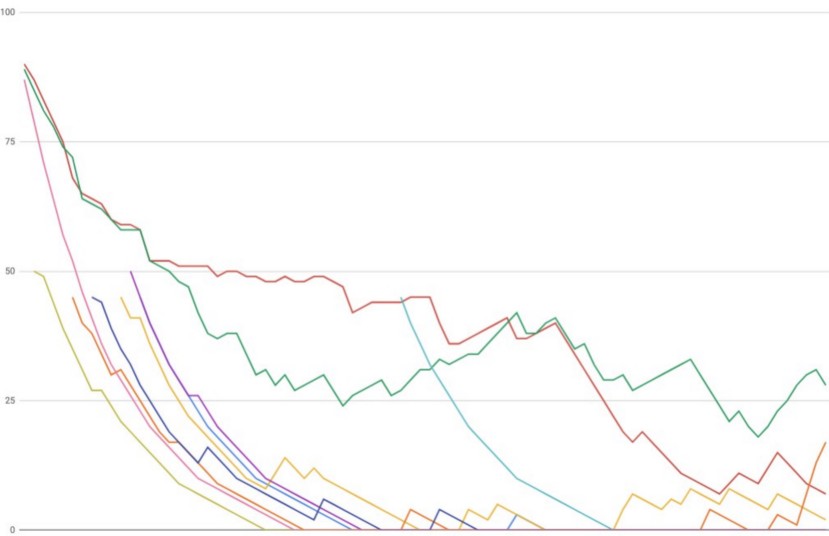

Figure 3: Temporal dynamics of reputation values for randomly selected 5 accounts from "black-lists" and 5 accounts from "whales" list. Horizontal axis corresponds to time period of 3 months from left to right, vertical axis indicates reputation value in range from 0.0 to 1.0, labeled on scale from 0 to 100 on the chart.

## 7    Conclusion

We have came up with generalized ontology capable to describe possible interactions in a wide range of online environments such as social networks, marketplaces and financial ecosystems including blockchains.

We have designed and implemented Reputation System available as part of the open source Aigents project at `https://github.com/aigents/aigents-java/blob/master/src/main/java/net/webstructor/peer/Reputationer.java`. As a part of the implementation, the temporal Aigents graph database has been successfully evaluated for the purpose of storage and processing of the reputation data.

We were able to extract data described by the ontology from the real world social network and financial ecosystem based on public blockchain and have successfully evaluated the performance of the computations against known reference data.

The entire framework is expected to be the foundation of any multi-agent AI framework, so the evolution of distributed multi-agent AI architecture and dynamics will be based on the organic reputation scores earned by the agents that are part of it.

ACKNOWLEDGMENTS

This work was supported by a grant for research centers, provided by the Analytical Center for the Government of the Russian Federation in accordance with the subsidy agreement (agreement identifier 000000D730324P540002) and the agreement with the Novosibirsk State University dated December 27, 2023 No. 70-2023-001318.

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
