# OpenReview forum: "Generalized Reputation Computation Ontology and Temporal Graph Architecture"
_mathai.club/MathAI/2025/Conference — MathAI 2025 Oral_

### Official Review · Reviewer_LKbh · 2025-02-25
**The article is good, but requires important revisions.**

**Rating:** 6
**Confidence:** 4

**Review:**

Strengths:

The paper considers a new type of consensus based on reputation. This area of ​​research is very important today. Building new blockchains and mechanisms for protecting them. The article describes all of this quite well.

Weaknesses:

As for The problem of reliable democratic governance, the authors need to describe more detailed an overview of solutions (and there are some) and how reputational consensus can help in this direction.
The topic of AI is not covered. It is necessary to create a chapter where the possibilities of using AI in this direction are shown.
Mathematical formulas (string 198, 201, 206 and etc) need to be described in more detail. It is not always clear where they come from.

---

### Official Review · Reviewer_UBtv · 2025-02-26
**The article is good, but requires important revisions.**

**Rating:** 7
**Confidence:** 4

**Review:**

Strengths:
In this paper authors give definition of a general purpose system, special cases of which are social networks and blockchains.
Paper suggest a way for calculating reputation for each person in the system. They developed and open-source code for reputation calculation for graphs.


Weaknesses:
It would be beneficial to add examples of existing system with reputations scores, such as online markets with scores of sellers, sites like habr.com/stackmathexchange.com, where only people with high reputation can perform some actions like creating a post, e.t.c.
Authors showed that calculated reputation of bad (blacklisted people and whales) declines. But is crucial to show that reputation of good (normal) account does not decline. Without this it is unclear if reputation can be used as score to separate unwanted accounts among normal ones. If authors demonstrate that their reputation can be used for determining bad/good accounts it is important to calculate accuracy/precision/recall/etc for this method on existing systems and compare metrics against other exiting methods.
Paper does not mention ways to use AI or integrate it into this system. Those should be added to the article.

---

### Official Review · Reviewer_qfFj · 2025-02-27
**A novel approach to address reputation in democratic governance without AI, security and scalability considerations**

**Rating:** 7
**Confidence:** 4

**Review:**

The work presents a sophisticated framework for computing and maintaining reputation in decentralized communities.
Below is an evaluation of its strengths, weaknesses along with future recommendations.

# Strengths
- The formulas are logically consistent, incorporating normalization, logarithmic transformations, and weighted averages to handle various aspects of reputation.
- The framework addresses real-world challenges such as skewed reputation distributions, decay rates, and granularity, making it potentially applicable to diverse environments like social networks, financial ecosystems, and marketplaces.
- References to experiments (Ethereum blockchain) suggest that the framework has been tested in real-world scenarios, enhancing its credibility.
- The integration of "Reputation Consensus", "Proof-of-Reputation," and "Reputation Mining" aligns with modern decentralized systems, showcasing forward-thinking design.

# Weaknesses
- The formulas assume positive ratings. If negative ratings are allowed, additional logic may be needed to handle them appropriately.
- The blending parameter and its determination are not fully explained, leaving a gap in understanding how aspect-specific weights are assigned.
- For large-scale systems, the computational complexity of these formulas could become an issue. Optimization techniques may be needed.
- While "Proof-of-Reputation" is mentioned, the work does not provide detailed mechanisms to mitigate manipulation and vulnerabilities.
- Lack of AI-based solutions sets questions how AI can be integrated in the framework.

# Recommendations
By incorporating concepts like "Proof-of-Reputation," the work aligns with the growing trend toward decentralized and blockchain-based systems, providing straightforward mathematical approach to its realization. The framework can be applied to various domains, including social networks, financial ecosystems, and marketplaces, making it versatile and potentially impactful.
However, fine-grained reputations and frequent updates could lead to high computational costs in large communities, so the focus on future development could investigate further computational cost and time scoping to discover the framework's potential for large-scale implementations. Additionally, detailed investigation of potential scenarios involving manipulation in technical and communal levels would be beneficial for further adoption where risk management and technical countermeasures are integrated. Finally, the future studies should include the influence and utilization of AI, along with use case scenarios.

# Conclusions
In summary, this paper provides technically compact approach for reputation-based proofing. The technical prowess and originality — along with potential impact — are evident despite the lack of solutions to address AI integration, scalability, and security.

---

### Comment · Reviewer_u513 · 2025-03-01
**Review Report: Generalized Reputation Computation Ontology and Temporal Graph Architecture**

1. Relevance of the Research Aim
The research addresses a critical challenge in decentralized governance: achieving fair, democratic consensus in communities and distributed systems. Current blockchain consensus mechanisms (e.g., Proof-of-Work (PoW), Proof-of-Stake (PoS)) perpetuate power imbalances by favoring computational resources or financial capital. The proposed Proof-of-Reputation (PoR) system aligns with emerging demands for equitable governance models, particularly as digital communities expand. By prioritizing earned reputation over brute force or wealth, the work directly responds to societal and technological needs for liquid democracy and manipulation-resistant evaluation frameworks.

2. Research Gap
Existing reputation systems lack robust mechanisms to:
•	2.1. Mitigate manipulation: Many platforms suffer from "reputation gaming," where actors exploit rating systems for undue influence.
•	2.2. Integrate temporal dynamics: Prior models often neglect the decay of historical reputation data, reducing responsiveness to recent behavior.
•	2.3. Generalize across domains: Most systems are siloed (e.g., social networks vs. marketplaces), limiting interoperability. The authors identify these gaps through critiques of PoW/PoS and fragmented reputation system designs, positioning PoR as a holistic alternative.

3. Scientific Novelty
Key innovations include:
•	3.1. Temporal weighted liquid rank algorithm: Integrates time-sensitive reputation decay, ensuring recent evaluations outweigh older ones.
•	3.2. Generalized ontology: Unifies reputation computation across social networks, financial ecosystems, and marketplaces via entities like Accounts, Smart Contracts, and Posts/Comments.
•	3.3. Incremental graph database design: Enables real-time reputation updates using temporal subgraphs, balancing computational efficiency and accuracy. These contributions advance reputation systems beyond static, domain-specific models.

4. Authors' Contribution
•	4.1. Conceptual framework: Developed PoR consensus as a democratic alternative to PoW/PoS.
•	4.2. Algorithmic design: Created formulas for differential reputation computation with logarithmic normalization to address nonlinear financial distributions.
•	4.3. Empirical validation: Tested the system on Steemit blockchain data, correlating computed reputations with manual "whale" and "blacklist" classifications.
•	4.4. Open-source implementation: Released tools for temporal graph storage and incremental recalculation.

5. Methodology
•	5.1.Hybrid approach: Combines theoretical modeling (e.g., ontology design) with empirical analysis of blockchain transactions and social interactions.
•	5.2. Algorithmic validation: Used real-world data to simulate reputation dynamics, demonstrating decay effects and resistance to manipulation.
•	5.3. Graph database implementation: Leveraged temporal subgraphs to manage large-scale financial and social data efficiently. While rigorous, the methodology could benefit from comparative studies against other consensus models (e.g., PoS in Ethereum).

6. Suggestions for Improvement
6.1.	Expand empirical validation: Test the system in diverse environments (e.g., Distributed Asynchronous Object Storage - DAOs, supply chains) to assess generalizability.
6.2.	Address scalability: Evaluate performance with ultra-large datasets to identify bottlenecks in the temporal graph architecture.
6.3.	Incorporate sybil-resistance: Integrate cryptographic techniques (e.g., zero-knowledge proofs) to prevent the creation of fake accounts, a noted limitation.
6.4.	Comparative analysis: Benchmark PoR against PoW/PoS in energy efficiency, transaction throughput, and decentralization metrics.
6.5.	Interdisciplinary collaboration: Partner with governance theorists to refine liquid democracy parameters (e.g., delegation thresholds).
Rating: 10: Good paper, accept

---

### Decision · Program_Chairs · 2025-03-08

**Decision:**

Accept (Oral)

**Comment:**

Your article has been accepted and you can give a talk on the article. All articles will be sorted by rating and within the available conference places one author from each article will be invited. If there are not enough places, then you will either have the opportunity to speak remotely or come at your own expense!